# Large Language Model Inference Acceleration Based on Hybrid Model Branch Prediction

Gaoxiang Duan [1,2], Jiajie Chen [1,2], Yueying Zhou [1,2], Xiaoying Zheng [1,2,*] and Yongxin Zhu [1,2,*]

1 Shanghai Advanced Research Institute, Chinese Academy of Sciences, Shanghai 201210, China; duangx@sari.ac.cn (G.D.); chenjiajie@sari.ac.cn (J.C.); zhouyueying@sari.ac.cn (Y.Z.)
2 University of Chinese Academy of Sciences, Beijing 100049, China
* Correspondence: zhengxy@sari.ac.cn (X.Z.); zhuyongxin@sari.ac.cn (Y.Z.)

**Abstract:** As the size of deep learning models continues to expand, the elongation of inference time has gradually evolved into a significant challenge to efficiency and practicality for autoregressive models. This work introduces a hybrid model acceleration strategy based on branch prediction, which accelerates autoregressive model inference without requiring retraining and ensures output consistency with the original model. Specifically, the algorithm employs two models with different parameter sizes aimed at the same task. The smaller model generates a series of potential tokens that are then parallelly validated by the larger model to determine their acceptability. By orchestrating the workflow of the large and small models through a branch-prediction strategy, the algorithm conceals the validation time of the larger model when predictions are successful, thereby accelerating inference. We propose a binomial distribution-based prediction function that blends theoretical principles with empirical evidence, specifically designed for the nuanced requirements of accelerating inference within a hybrid model framework. The entire algorithm was designed and implemented on the llama model for text generation and translation tasks. The experimental results indicate significant improvements. The proposed algorithm achieves a $1.2\times$ to $3.4\times$ increase in inference speed compared to the original model, consistently outperforming the speculative sampling inference acceleration algorithm.

**Keywords:** large language model; auto regressive model; branch prediction; decoder; inference





## 1. Introduction

Autoregressive large language models (LLMs) based on the transformer [1] architecture have revolutionized the field of natural language processing (NLP). The scale advantage of these models has brought significant performance improvements to language generation tasks, contributing to groundbreaking improvements across a wide array of applications, from automated content creation to real-time translation services [2–4]. The pivotal role of LLMs in advancing the frontiers of artificial intelligence and their utility in processing and understanding complex language patterns cannot be overstated.

However, the evolution of these models introduces a critical challenge: their increased complexity leads to longer inference times. As the size and sophistication of these models grow, so does the computational cost required to generate outputs, leading to inefficiencies that limit their practical applicability, especially in scenarios demanding real-time responses. This burgeoning problem has catalyzed a flurry of research aimed at devising strategies to accelerate inference without sacrificing the models' output quality, a crucial endeavor for sustaining the momentum of innovation in the field of NLP.

Efforts to mitigate the inference bottleneck of LLMs in NLP have concentrated on two principal strategies. Firstly, optimizations of transformer model architectures aimed at enhancing decoder efficiency have been pursued to diminish the overall runtime [4–7]. Such optimizations, however, often necessitate extensive retraining, incurring significant

computational costs and negating the benefits of existing pretrained models. Secondly, hybrid model strategies have been explored, where smaller models assist in the initial generation task, potentially accelerating the process. Despite their promise, these strategies are limited by the inherent output quality discrepancies between different model sizes, which can adversely affect overall performance. Google's 2023 proposal of a hybrid model strategy called speculative sampling [8], which employs small models for draft generation and large models for draft validation, marked a significant departure by decoupling the sequential execution of large models in the generation process, thereby achieving notable inference time reductions. However, the emergence of workflow blockages during the validation phase has been identified as a critical performance bottleneck. In response, building upon the principles of speculative sampling, this study introduces an advanced fast inference algorithm that incorporates branch-prediction techniques. Specifically designed to curtail the inference time of large autoregressive models without sacrificing the output quality, this refined approach addresses the limitations inherent in prior methods. By leveraging and extending the speculative sampling framework, our proposed solution significantly enhances the inference efficiency within the NLP domain, offering a sophisticated advancement over existing strategies. To address the blocking issue in the validation process of hybrid model inference, this paper introduces branch-prediction technology and proposes a fast inference algorithm for hybrid models based on branch prediction. By immediately predicting the possible starting positions with the help of a predefined prediction function after the small model execution is completed, the parallel execution of the small model and large model validation is achieved. This method significantly reduces the inference time.

The contributions of this paper include the following:

1. Firstly, we propose a hybrid model acceleration inference method based on branch prediction. By using branch prediction, the validation time during hybrid model inference would be reduced, which significantly reduces the inference time.
2. Secondly, we construct a branch-prediction function based on the binomial distribution assumption to fit the empirical distribution, further accelerating the inference speed.
3. Lastly, through experiments, it is demonstrated that the proposed algorithm achieves better acceleration effects in tasks of generating combinations of models of different scales.

The remainder of this paper is organized as follows: We discuss the related works in Section 2. The whole method is described in Section 4, and the detailed analysis is introduced in Section 5. We present the experimental results in Section 6. The conclusion is drawn in Section 7.

## 2. Related Work

There has been extensive research on efficient inference for large models [7,9,10]. Previous work on efficient inference for LLMs has mainly focused on two aspects: accelerating the inference speed of the single-round decoder and the entire inference process. One category of techniques attempts to reduce the computational complexity by changing the model structure to accelerate single-model inference, such as distillation [11], sparsification [12], quantization [13], and architectural modifications [14]. These methods have accelerated the model-inference process but require the model to be retrained, which has the drawback of not being able to use existing pretrained models. Moreover, due to modifications in the model structure, the final output cannot maintain consistency with the original target model, resulting in considerable performance loss. Knowledge distillation transfers knowledge from large models to smaller ones to reduce computational load, but this process often requires additional training for the smaller models. Model sparsification techniques reduce the number of non-zero parameters in the model to lower storage and computational demands, which may lead to a decrease in the model expressiveness. Quantization methods compress the model by reducing the bit width of the model parameters, which

can significantly reduce computational resources while maintaining performance but may introduce quantization errors. Architectural modifications optimize inference speed by redesigning the model structure, such as introducing more efficient attention mechanisms, but may require complete model restructuring.

Another category, adaptive computation methods, attempts to accelerate the entire inference process. These methods are based on an important observation: during the inference-generation process, some simple generation tasks can be approximated by smaller models. Prior work has made related attempts [4,7,13,15]. Recently, Han et al. [16] proposed an adaptive computation method that adapts the computational effort to the difficulty of the problem. Sukhbaatar et al. [17] accelerated inference by having a small model handle some simple tasks in the output. However, these methods also cannot maintain consistent output with the original target model. In 2023, the Google team, led by Leviathan et al. [8], proposed a speculative sampling-based hybrid model adaptive inference method, using a small model as the approximate model. By generating drafts through the approximate small model and validating with the target large model, the time dependency in the autoregressive model-inference process is decoupled, allowing the inference process of the large model to be parallelized. The speculative sampling-based hybrid model acceleration inference algorithm effectively speeds up the inference time and has been applied in actual large language model inference scenarios. The comparison between different acceleration methods is listed in Table 1.

However, in the validation process of the large model, the generation by the small model must wait for the validation by the target large model to finish before restarting, making workflow blocking the main bottleneck in the inference speed of hybrid model algorithms.

**Table 1.** Comparison of different acceleration algorithms.

| Acceleration Algorithms | Acceleration Type | Need Retrain | Align Original Output |
|---|---|---|---|
| distillation [11] | | Yes | No |
| sparsification [12] | single-round decoder | Yes | No |
| quantization [13] | | Yes | No |
| architectural modifications [14] | | Yes | No |
| confident adaptive transformers [17] | | Yes | No |
| fast transformer decoding [18] | entire inference process | Yes | No |
| speculative sampling [8] | | No | Yes |

## 3. Prior Knowledge

Enhancing the speed of inference for modern LLMs has become an unavoidable challenge. Traditional inference methods are confronted with substantial computational resource consumption, which hinders the realization of real-time inference demands. Recently, speculative sampling [8] has been proposed as an effective method to accelerate inference and has demonstrated superior performance in various studies. However, despite the achievements in accelerating inference, the potential for efficiency improvement in speculative sampling has not been fully explored. In light of this, our study aims to explore the application of branch prediction, a technology widely used in computer architecture, in speculative sampling to further enhance the efficiency of hybrid model inference acceleration algorithms. By integrating the core concepts from both domains, we aspire to offer a new perspective and methodology for accelerating inference in LLMs. This section will introduce the fundamental principles and application backgrounds of speculative sampling and branch prediction, laying the groundwork for understanding the contributions of our work.

### 3.1. Speculative Sampling

Before delving into our research, it is crucial to understand the core idea and implementation of speculative sampling. Speculative sampling is a method designed to accelerate the inference process of large language models. It is predicated on the assumption that

not all steps in the inference process of a large autoregressive language model equally impact the quality of the final output. Therefore, by predicting which steps have minimal impact on the output quality and using smaller, faster models for inference at these steps, the inference process can be significantly accelerated without substantially sacrificing the output quality. The implementation of speculative sampling involves two key components: a smaller, faster model $M_s$ with $s(x|x < t)$ being the distribution of the small model $M_s$ based on the prefix $x < t$ and the original, larger model $M_b$ with $b(x|x < t)$ being the distribution of the large model $M_b$ based on the prefix $x < t$. The whole inference process is listed as follows:

1.  The smaller model is used to quickly generate a series of inference output drafts $n \in \mathbb{Z}^+$ characters (tokens) $\{x_i|x_i \sim s(x), 1 \leq i \leq n\}$.
2.  These drafts are then verified and, if necessary, corrected by the original large model. In detail, the number of accepted drafts $\delta$ is defined as

$$\delta = min(i - 1 \mid 1 \leq i \leq n, \varepsilon_i > \frac{b_i(x)}{s_i(x)} \cup n) \tag{1}$$

where $\varepsilon_i \sim U(0,1)$.
3.  After validation, adjust the result:

$$\begin{cases} \{x_i \mid 1 \leq i \leq \delta\}, n = \delta \\ \{x_1, ... x_\delta, \alpha\}, n > \delta \end{cases} \tag{2}$$

where $\alpha$ is defined to ensure the final output is just like the sample from the large model $x \sim b(x)$:

$$\alpha \sim norm(max(0, b(x|x < t) - s(x|x < t))) \tag{3}$$

The essence of this approach lies in the ability of the smaller model to predict and skip over parts that have little impact on the final output, concentrating computational resources on inference steps critical to the output quality. Figure 1 illustrates an example of how speculative sampling accelerates the entire generation process.

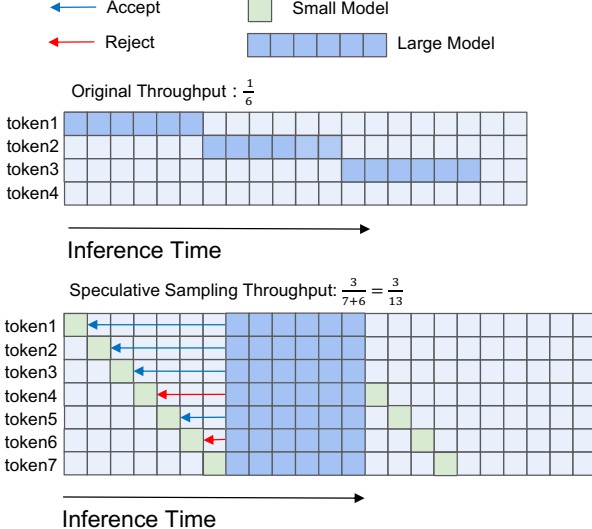

**Figure 1.** The schematic diagram of speculative sampling compared with original generation. In the original output scenario, due to the autoregressive nature of the model, the generation of each subsequent word must commence only after the preceding word has been finalized, with the large model requiring six units of time for a single execution. However, under the speculative sampling approach, this sequential execution is limited to the generation phase by a faster, smaller model, which

requires only one unit of time for execution. The larger model's validation process can then be conducted in parallel, based on the content already produced by the smaller model, thereby accelerating inference speed and increasing throughput by efficiently utilizing the time differential between the small and large models.

### *3.2. Branch Prediction*

Branch prediction is a pivotal technique in computer architecture, aimed at enhancing the efficiency of executing instruction sequences by modern processors. The core idea revolves around predicting the behavior of conditional branches such as if–else statements in a program to reduce delays caused by waiting for branch decisions. In processor design, branch prediction allows the processor to preload and execute instructions predicted as the subsequent steps before the actual branch outcome is determined. When predictions are accurate, this significantly improves the execution efficiency.

The key to implementing branch prediction lies in the design of prediction algorithms, which must predict the execution of conditional branches in a program quickly and accurately. The most basic branch-prediction strategies include static prediction and dynamic prediction:

- Static prediction typically relies on simple rules, such as always predicting that a branch will go in a specific direction (for example, always predicting that a loop will continue).
- Dynamic prediction, on the other hand, depends on information collected at runtime, predicting future branch decisions based on historical branch behavior. This category includes a range of complex algorithms, such as Two-Level Adaptive Training and History Table-Based Prediction.

## 4. Methods

This paper proposes a hybrid model acceleration inference algorithm based on branch prediction, with the schematic diagram shown in Figure 2. Section 3.1 introduces the workflow of the hybrid model generation algorithm based on branch prediction. Section 3.2 discusses the design of the prediction function in the acceleration algorithm.

### *4.1. Hybrid Model Inference Acceleration Algorithm Based on Branch Prediction*

The inference acceleration algorithm based on branch prediction proposed in this paper divides the overall generation task into multiple rounds, each comprising the following four steps. The algorithm flowchart is shown in Figure 1.

Let $M_b$ be the target large model to be accelerated, with $b(x|x < t)$ being the distribution of the target model $M_b$ based on the prefix $x < t$. The time required for a single inference of the target large model $M_b$ is denoted as $T$. We define $M_s$ as a smaller-scale approximate model tasked with the same inference job, and the probability distribution of this model, conditioned on the prefix $x < t$, is referred to as $s(x|x < t)$. The time needed for a single inference of the approximate model $M_s$ is $t$, which is $T$ divided by $k(T/k)$.

1. Small Model Draft Generation: Initially, the small model $M_s$ generates drafts of $n \in \mathbb{Z}^+$ tokens $\{x_i \mid x_i \sim s(x), 1 \le i \le n\}$, which could be the potential generation output after the validation of the large model. The objective of this phase is to swiftly produce preliminary inference outcomes using the small model. These initial results then enable the large model to overcome sequence limitations by utilizing these drafts as a foundation.
2. Branch Prediction: The prediction function $\phi(x)$ is employed to predict the number of acceptable drafts $\gamma$. The first $\gamma$ tokens are accepted, and $\gamma + 1$ is set as the starting point for the next round of small model $M_s$ inference, followed by the commencement of the next round generation. By introducing branch prediction, we optimized the scheduling of the entire inference process. In the subsequent steps, we will observe that the total inference time is reduced when predictions are accurate.

3. Large Model Validation: Concurrently with the generation of the next round of drafts by the small model, the large model $M_b$ parallelly evaluates all draft tokens from $M_s$ of this round and their respective probabilities to determine the actual number of accepted $\delta \in \mathbb{Z}$. If $\epsilon \geq b(x_i)/s(x_i)$, it signifies that the output result of the target model at this position is consistent with that of the approximate model, accepting the token generated by the approximate model at the $i$th position of this round; otherwise, it is considered a failure in draft generation by the approximate model at this position. The longest consecutive number of accepted tokens starting from the initial position is counted as the real number of accepted $\delta$ for this round:

$$\delta = min(i - 1 \mid 1 \leq i \leq n, \varepsilon_i > \frac{b_i(x)}{s_i(x)} \cup n) \tag{4}$$

The validation step will determine the actual acceptable draft length, similar to speculative sampling. The retained tokens will ensure consistency with the results directly sampled from the large model.

4. Branch-Prediction Result Check: After the validation by the large model $M_b$ is completed, the prediction for this round is checked:

$$\Theta(\delta, \gamma) = T - (\delta - \gamma)t \tag{5}$$

If $\Theta(\delta, \gamma) > 0$, the prediction is deemed successful, and the small model proceeds to the next round of draft generation. Otherwise, in the event of prediction failure, the process reverts to the verification point and restarts the generation process. The determination of whether a prediction is successful is based on the comparison between the predicted values and the actual values. Whether the prediction is successful or not also decides if the inference process can be accelerated.

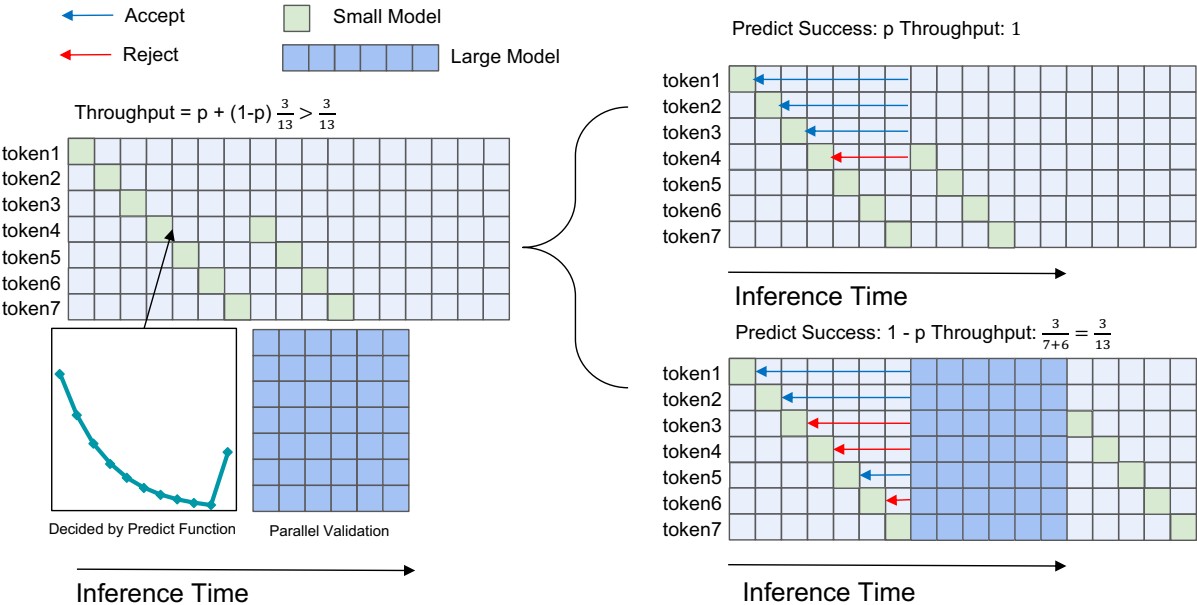

**Figure 2.** The schematic diagram of hybrid model inference based on branch prediction. For a single execution, the large model requires six units of time, whereas the small model requires one unit of time. Utilizing a predictive function allows for the immediate determination of a new starting point upon the completion of small model generation, enabling parallel validation by the large model. In scenarios where the prediction is successful, the validation time of the large model is effectively concealed, rendering it negligible from a runtime perspective. Conversely, in cases of prediction failure, the runtime remains consistent with that of speculative sampling. Considering both successful and unsuccessful predictions, our approach invariably enhances inference speed.

The steps mentioned above constitute the complete algorithm implementation. The method proposed in this article is built upon the foundation of speculative sampling schemes, further enhancing efficiency. It completely conceals the generation time of the large model when predictions are successful and maintains the same generation speed as speculative sampling when predictions fail. This means that the introduction of a branch-prediction strategy will result in a speed increase, even with a poor prediction accuracy rate.

### 4.2. Design of Prediction Function for Acceleration

Before the inference starts, based on the determined $M_b$, $M_s$, and the completion number $n$ generated each round, obtain the probability density function $\phi(x)$ for the number of tokens that can be accepted each round. The parameters $a$ and $b$ are determined by the choice of models $M_b$, $M_s$, and $n$. The probability density function, as the prediction function in branch prediction, is determined before the inference starts, and the predicted value $\gamma \sim \phi(x)$ is determined by sampling from the probability density function $\phi(x)$ with $1 \leq \gamma \leq n$.

## 5. Analysis

### 5.1. Impact of Branch-Prediction Strategy on Throughput

This paper measures the inference speed of different strategies using the number of tokens generated per unit time, which is the throughput $\omega$. Since the proposed acceleration method aims to align with the target model output, a higher throughput indicates less total inference time used. Here, we analyze and compare the throughput of our acceleration algorithm with the standard inference process and the fast inference algorithm based on speculative sampling [8]. The throughput of the target large model is

$$\omega_o = 1/T \tag{6}$$

The throughput of the fast inference algorithm based on speculative sampling is

$$\omega_s = \frac{\beta(n)}{nt + T} \tag{7}$$

where $\beta$ is the average number of tokens that can be accepted in each round of n generated tokens. A larger $\beta$ indicates a more significant acceleration effect. The throughput of the inference acceleration algorithm proposed in this paper is

$$\omega_b = p \times \frac{\beta(n)}{nt + T} + (1 - p) \times \frac{\beta(n)}{nt + (\delta - \gamma)t} \tag{8}$$

where $p$ represents the probability of correct branch prediction. The impact of the branch-prediction strategy on time depends on the outcome of the branch prediction. When the prediction result is correct, which is the check function $\Theta(\delta, \gamma) \geq 0$, the round of generation saves the time needed for large model validation. When the prediction result is incorrect, which is the check function $\Theta(\delta, \gamma) < 0$, a recall operation is performed to return to the last generation end point and choose the correct position to restart. At this time, the time spent is the same as the algorithm without branch prediction [8]. Based on the above analysis, the fast inference algorithm based on branch prediction proposed in this paper compared to algorithm [8]:

$$\omega_b = p \times \frac{\beta(n)}{nt + T} + (1 - p) \times \frac{\beta(n)}{nt + (\delta - \gamma)t} \geq \frac{\beta(n)}{nt + T} = \omega_s \tag{9}$$

Overall, compared to the fast inference algorithm based on speculative sampling, choosing the acceleration algorithm of this paper will reduce the inference time. The degree

of acceleration depends on the effectiveness of the prediction function, i.e., the size of the probability $p$ of correct branch prediction.

### 5.2. Prediction Function Optimization

5.2.1. Prediction Function Based on Independent and Identically Distributed (IID) Theory Analysis

The choice of the prediction function directly affects our acceleration effect. Here, this paper follows the assumption of speculative sampling [8]; that is, each time the output of the approximate model and the target model is consistent is an independent and identically distributed event. At this time, the probability $p(k)$ of the first $k$ words being correctly output is

$$p(k) = \alpha^k(1 - \alpha) \tag{10}$$

where $\alpha \in [0, 1]$ is the probability that the output of the approximate model is consistent with the target model given the same prefix. Based on the IID assumption and according to the probability density function, the distribution function regarding the correct times has the following characteristics:

- The probability density distribution function is a monotonically decreasing convex function.
- There is a significant increase in probability at $n$ compared to the adjacent probability before it. The probability at $n$ should be the sum of all probabilities greater than or equal to $n$; that is:

$$p(n) = 1 - \sum_{i=1}^{n-1} \alpha^i(1 - \alpha) = 1 - \alpha + \alpha^n \tag{11}$$

Figure 3 shows the possible probability distribution functions under different $\alpha$ when the single-round generation count $n = 15$ is based on the I.I.D. assumption.

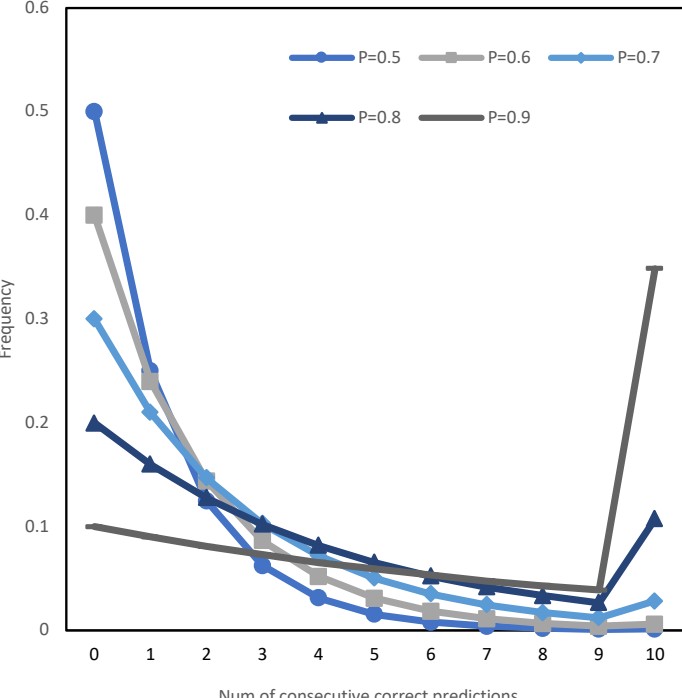

**Figure 3.** Probability distribution of the number of correct predictions based on the I.I.D. assumption.

5.2.2. Prediction Function Based on Empirical Distribution Experimental Analysis

This paper uses the llama model 15 M as the approximate model and 42 M as the target model, counting the results of 500 text-generation-task experiments. The statistics of the consecutive correct prediction frequency for single-round counts of $n = 10$ and $n = 15$ are shown in Figure 4.

From the figure, it can be observed that the frequency distribution of correct predictions has the following characteristics:

- The continuous approximation function of the frequency distribution is a monotonically decreasing convex function.
- There is a significant abnormal increase in the frequency distribution at the end $n$ compared to the adjacent frequency data before it.

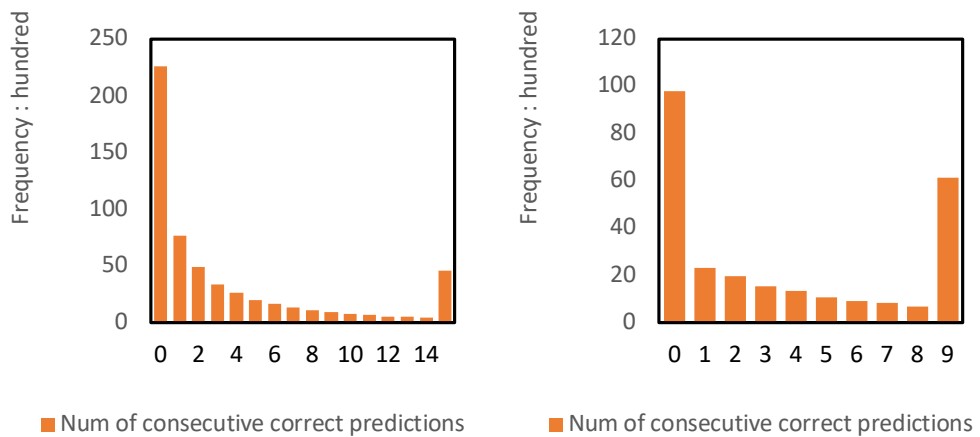

**Figure 4.** Frequency distribution of the number of correct predictions.

5.2.3. Determination of the Prediction Function

The theoretical probability distribution function and the frequency distribution obtained from the empirical distribution have similar distribution function characteristics: both are monotonically decreasing convex functions, and there is a significant increase at $n$ compared to the adjacent data before it. The similarity of the frequency distribution to the frequency distribution to some extent demonstrates the reliability of the theoretical assumption in 4.2. Further, the I.I.D. assumption is a simple approximation of reality; therefore, this paper adds two correction terms to more closely fit the final empirical probability density function. The final prediction function $\phi(x)$ is obtained as

$$\phi(x) = \begin{cases} ap^x(1-p) + c, & \text{for } 0 \le x \le n - 1 \\ 1 - \sum_{i=0}^{n-1} [ap^i(1-p) + c], & \text{for } x = n \end{cases} \tag{12}$$

where $a$ and $c$ are parameters of the correction terms.

According to the prediction function, the impact of the number of drafts generated per round is shown in Figures 5–7.

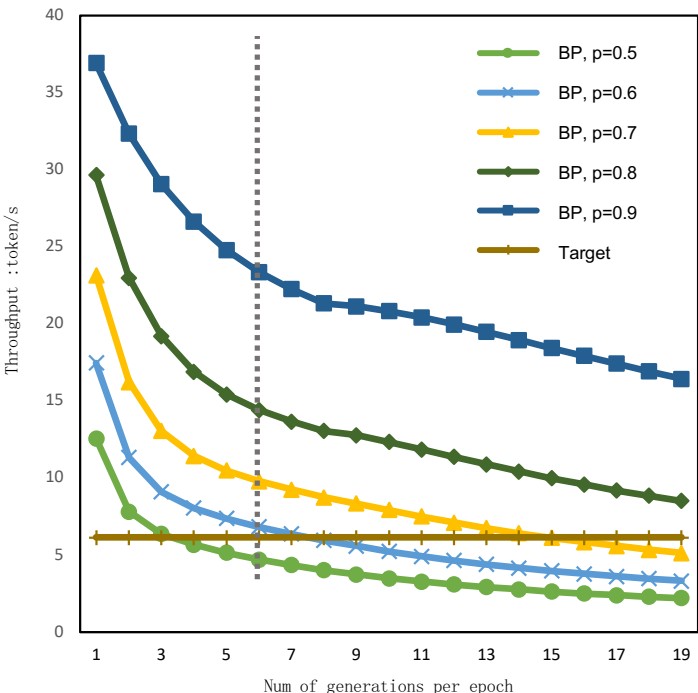

**Figure 5.** The impact of the number of drafts generated per round by our work on throughput. BP represents the method with branch-prediction strategy. The vertical bar represents the minimum number of tokens that can be generated in a single round. Higher is better.

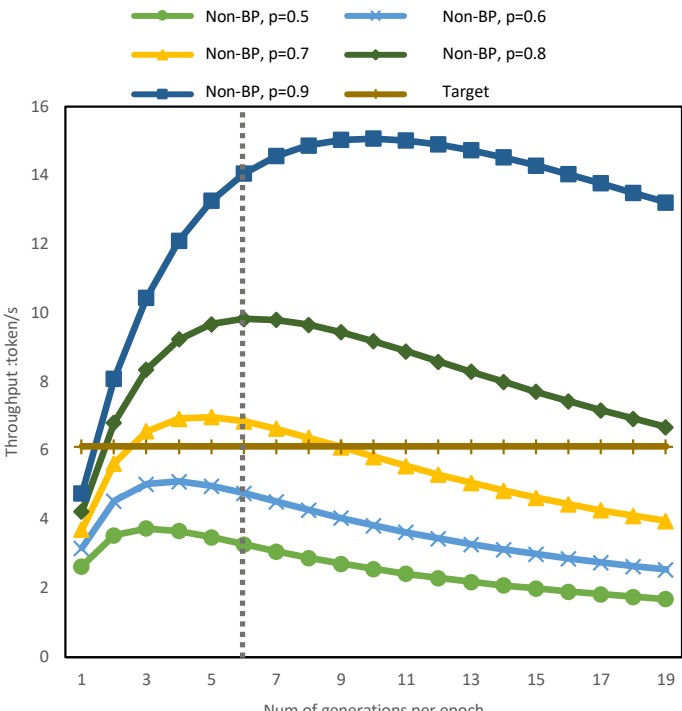

**Figure 6.** The impact of the number of drafts generated per round by the speculative sampling strategy on throughput. Non-BP represents without branch prediction. The vertical bar represents the minimum number of tokens that can be generated in a single round. Higher is better.

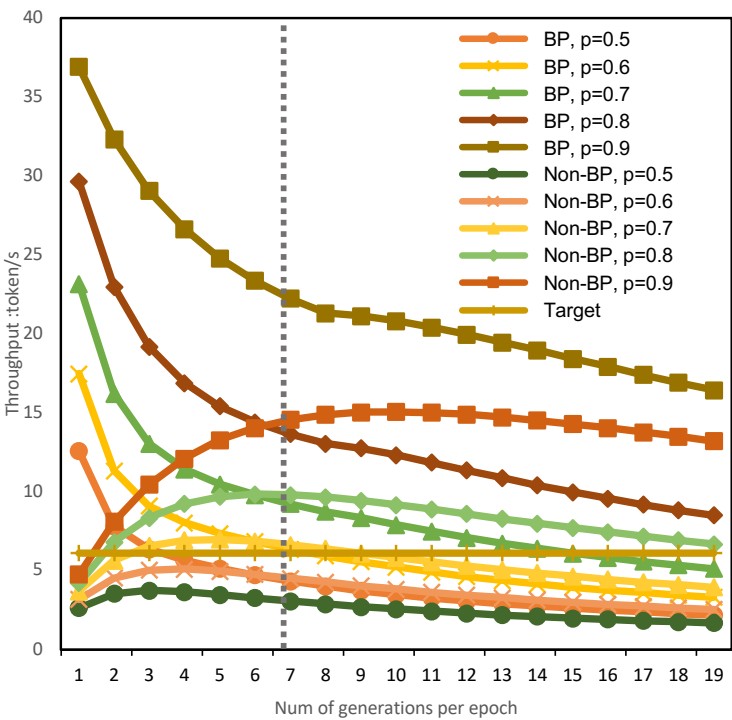

**Figure 7.** The comparison of the impact of the number of drafts generated per round on throughput. The vertical bar signifies the minimum number of tokens that can be generated in a single epoch. A higher value is preferable. Given the same probability settings, the throughput of the method employing branch prediction consistently surpasses that of the method without branch prediction.

## 6. Experiments

### 6.1. Experiments Setup

All the models were thoroughly pretrained on diverse text corpora prior to the experiments. Both our approach and the benchmark speculative sampling algorithm ensure that the outputs perfectly aligned with the target model, resulting in consistent outputs across the three models compared. Our experiment focuses solely on the throughput of each model.

The models used in this experiment are based on the llama [19] architecture, including four different scales of models with parameter sizes of 260 K, 15 M, 42 M, and 110 M. Table 2 outlines the parameter configurations of the llama models utilized in our experiments. In the experimental setup, the model temperature is set to 0.8, with top-p sampling at 0.9. The use of top-p sampling ensures that tokens with minuscule probabilities are not sampled, thus contributing to the efficiency and relevance of the generated text.

**Table 2.** Parameters of models at different scales.

| Model | Dimension | Number of Attention Heads | Layers | Parameters |
| --- | --- | --- | --- | --- |
| 260 K | 64 | 8 | 5 | 260 K |
| 15 M | 288 | 6 | 6 | 15 M |
| 42 M | 512 | 8 | 8 | 42 M |
| 110 M | 768 | 12 | 12 | 110 M |

### 6.2. The Impact of Branch-Prediction Algorithm across Diverse Model Scales

This section validates the proposed branch-prediction acceleration algorithm's effectiveness across a spectrum of model scales, ranging from small to large configurations. For the text-generation task, this study employs the TinyStories dataset [20]. Meanwhile, the translation tasks utilize the WMT 2018 English–French dataset [21].

This experiment utilizes throughput as a quantitative measure to evaluate the models' generation speed:

$$Throughput = \frac{\#generated\ tokens}{time} = \frac{1}{computing\ time} \qquad (13)$$

This experiment compared the computation times of different algorithms under the same model scale configuration. The comparative results of the computation time for the text-generation task are presented in Figure 8, and those for the translation task are also shown in Figure 9. Furthermore, the detailed throughput data for the text-generation task are listed in Table 3 and for the translation task in Table 4.

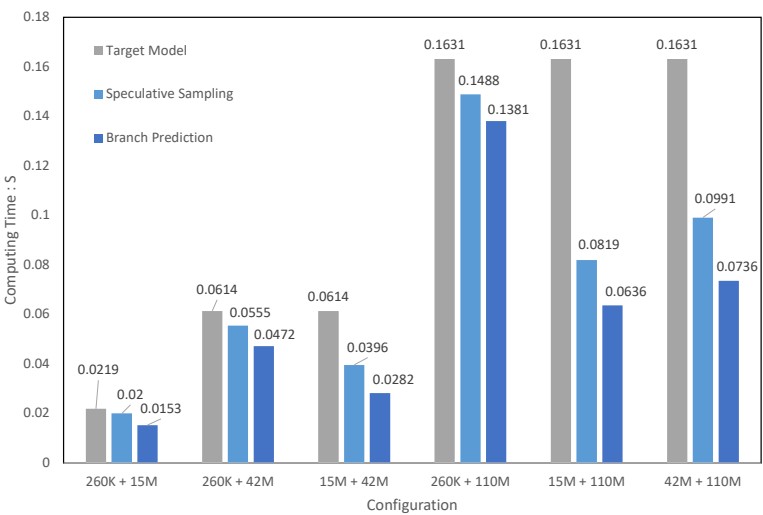

**Figure 8.** Comparative analysis of computing times across various algorithms for text-generation tasks under different model configurations: lower times indicate better performance.

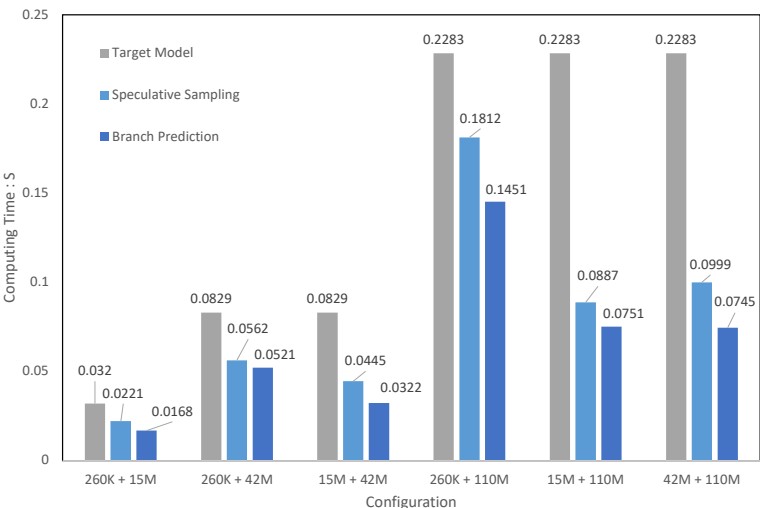

**Figure 9.** Comparative analysis of computing times across various algorithms for translation tasks under different model configurations: lower times indicate better performance.

The experimental results indicate that the branch-prediction algorithm proposed in this paper, while maintaining the output consistency with the target model, achieves a higher throughput than both the target large model and the speculative sampling algorithm proposed by the Google team [8] across different tasks and model configurations.

In text-generation tasks, the branch-prediction algorithm demonstrates a significant throughput improvement compared to both the target large model and the speculative

sampling algorithm proposed by the Google team. Specifically, within the setup using a 260 K small model coupled with a 15 M target model, the branch-prediction algorithm reached a throughput of 65.33 tokens/s. This is 1.3 times the throughput of the speculative sampling algorithm at 50.12 tokens/s and 1.4 times higher than the target model's 45.59 tokens/s. Remarkably, in the setup of 15 M plus 110 M, the branch-prediction algorithm achieved a throughput of 15.73 tokens/s, which is 2.6 times the throughput of the target large model at 6.13 tokens/s and substantially exceeds the speculative sampling algorithm's 12.21 tokens/s. Table 5 demonstrates our approach in a text-generation task, achieving the shortest computation time while maintaining a consistent content output.

In translation tasks, the branch-prediction algorithm continues to demonstrate its superior efficacy. For instance, within the framework utilizing a 260 K small model in conjunction with a 15 M target model, the branch-prediction algorithm attained a throughput of 59.53 tokens/s. This performance is approximately 31% higher than the speculative sampling algorithm's 45.33 tokens/s and nearly 1.9 times the throughput of the target model at 31.29 tokens/s. Even more impressive, in a configuration combining a 42 M small model with a 110 M target model, the branch-prediction algorithm realized a throughput of 13.42 tokens/s. This rate is 1.34 times the throughput achieved by the speculative sampling algorithm at 10.01 tokens/s and 3.4 times greater than the target model's 4.38 tokens/s. These outcomes underscore the branch-prediction algorithm's remarkable acceleration capability, especially in configurations involving larger models.

These results clearly demonstrate that the branch-prediction algorithm, while maintaining output consistency with the target model, offers substantial performance improvements in accelerating autoregressive model inference compared to the existing speculative sampling algorithm. The achievement of such acceleration effects, especially in translation and text-generation tasks with high-throughput demands, validates the effectiveness and superiority of the branch-prediction algorithm in practical applications.

Additionally, the scale of the approximate model does not linearly correlate with the acceleration effect. In the text-generation task targeting the 110 M model, the 15 M approximate model achieved the highest acceleration effect of 2.6 times. In the translation task, targeting the same 110 M model, the 42 M approximate model reached the highest acceleration effect of 3.4 times. In practical applications, the appropriate approximate model should be selected based on the specific task scenario and target model.

**Table 3.** Comparison of inference throughput under different hybrid model configurations in text-generation tasks.

| Hybrid Model Configuration | Small Model Throughput (tokens/s) | Target Model Throughput (tokens/s) | Speculative Sampling Throughput (tokens/s) | Branch-Prediction Throughput (tokens/s) |
|---|---|---|---|---|
| 260 K + 15 M | 2400 | 45.59 | 50.12 | 65.33 (1.4×) |
| 260 K + 42 M | 2400 | 16.29 | 18.02 | 21.18 (1.3×) |
| 15 M + 42 M | 45.59 | 16.29 | 25.23 | 35.51 (2.2×) |
| 260 K + 110 M | 2400 | 6.13 | 6.72 | 7.24 (1.2×) |
| 15 M + 110 M | 45.59 | 6.13 | 12.21 | 15.73 (2.6×) |
| 42 M + 110 M | 16.29 | 6.13 | 10.09 | 13.58 (2.2×) |

**Table 4.** Comparison of inference throughput under different hybrid model configurations in translation tasks.

| Hybrid Model Configuration | Small Model Throughput (tokens/s) | Target Model Throughput (tokens/s) | Speculative Sampling Throughput (tokens/s) | Branch-Prediction Throughput (tokens/s) |
|---|---|---|---|---|
| 260 K + 15 M | 1852 | 31.29 | 45.33 | 59.53 (1.9×) |
| 260 K + 42 M | 1852 | 12.06 | 17.79 | 19.21 (1.6×) |
| 15 M + 42 M | 31.29 | 12.06 | 22.48 | 31.01 (2.6×) |
| 260 K + 110 M | 1852 | 4.38 | 5.52 | 6.89 (1.6×) |
| 15 M + 110 M | 31.29 | 4.38 | 11.28 | 13.31 (3.2×) |
| 42 M + 110 M | 12.06 | 4.38 | 10.01 | 13.42 (3.4×) |

**Table 5.** Performance comparison in text-generation task between our work, speculative sampling, and the origin method.

| Method | Origin (110 M) | Speculative Sampling (110M + 42 M) | Our Work (110 M + 42 M) |
|---|---|---|---|
| Content | One day, Lily met a Shoggoth. He was very shy, but was also very generous. Lily said "Hello Shoggy! Can I be your friend?" Shoggy was happy to have a friend and said "Yes, let's explore the universe together!" So they set off on a journey to explore the universe. As they travelled, Shoggy was happy to explain to Lily about all the wonderful things in the universe. At the end of the day, Lily and Shoggy had gathered lots of wonderful things from the universe, and they both felt very proud. They promised to explore the universe as one big pair and to never stop being generous to each other. | One day, Lily met a Shoggoth. He was very shy, but was also very generous. Lily said "Hello Shoggy! Can I be your friend?" Shoggy was happy to have a friend and said "Yes, let's explore the universe together!" So they set off on a journey to explore the universe. As they travelled, Shoggy was happy to explain to Lily about all the wonderful things in the universe. At the end of the day, Lily and Shoggy had gathered lots of wonderful things from the universe, and they both felt very proud. They promised to explore the universe as one big pair and to never stop being generous to each other. | One day, Lily met a Shoggoth. He was very shy, but was also very generous. Lily said "Hello Shoggy! Can I be your friend?" Shoggy was happy to have a friend and said "Yes, let's explore the universe together!" So they set off on a journey to explore the universe. As they travelled, Shoggy was happy to explain to Lily about all the wonderful things in the universe. At the end of the day, Lily and Shoggy had gathered lots of wonderful things from the universe, and they both felt very proud. They promised to explore the universe as one big pair and to never stop being generous to each other. |
| Total Computing Time: s | 18.10 | 10.99 | 8.10 |

### 6.3. Ablation Experiment on the Impact of Single-Round Token Generation Quantity on Acceleration Effect

This section analyzes the impact of the number of tokens generated by the approximate model in a single round on the acceleration effect, with the 110 M model as the target model and the 15 M model as the approximate model in the text-generation experiment. The minimum value of n must satisfy the condition that the single-round draft time is greater than the target model's single verification time:

$$n \geq \lceil T/t \rceil \tag{14}$$

In this experiment, the test range is chosen as $n \in \mathbb{Z}^+, 8 \leq n \leq 20$. The experimental results are shown in Figure 10. From Figure 10, it can be seen that the number of tokens generated in a single round, n, is negatively correlated with the acceleration effect. At $n = 8$, we observe the optimal acceleration effect, achieving an inference speed of 15.73 tokens/s, representing a 2.6-fold increase over the target model's acceleration ratio. Conversely, at $n = 20$, the acceleration effect diminishes to its lowest, with an inference speed of 7.11 tokens/s, marking only a 1.2-fold improvement in the acceleration ratio relative to the target model. The impact of the number of tokens generated by the approximate model in a single round on the acceleration effect is consistent with the prediction in Section 5.

### 6.4. Extreme Trade-Off Strategy: Exploring Exhaustive Methods

Further, this section presents a more extreme trade-off strategy to explore the limit of time-friendly algorithms. After each execution of the small model, instead of branch prediction, all possible cases are executed in parallel, while the large model performs validation in parallel. After the next round of the small model execution is completed, the validation work of the large model has also been finished. Choose the correct path among the all-parallel small models verified as correct to continue execution and repeat the above operation. The comparison diagram is shown in Figure 11.

The exhaustive method can completely hide the validation time of the large model from a temporal perspective, but since it requires the parallel execution of n small models, it also significantly increases the use of computing resources. This section chose the 15 M and 110 M models for the text-generation-task experiment. The experimental results are shown in Figure 12. From the figure, it can be seen that the strategy of using the exhaustive

method reaches the highest inference speed of 20.01 tokens/s. However, it also increases up to eight times the computing resources of the approximate model. As n increases, the additional amount of the approximate model also continuously increases.

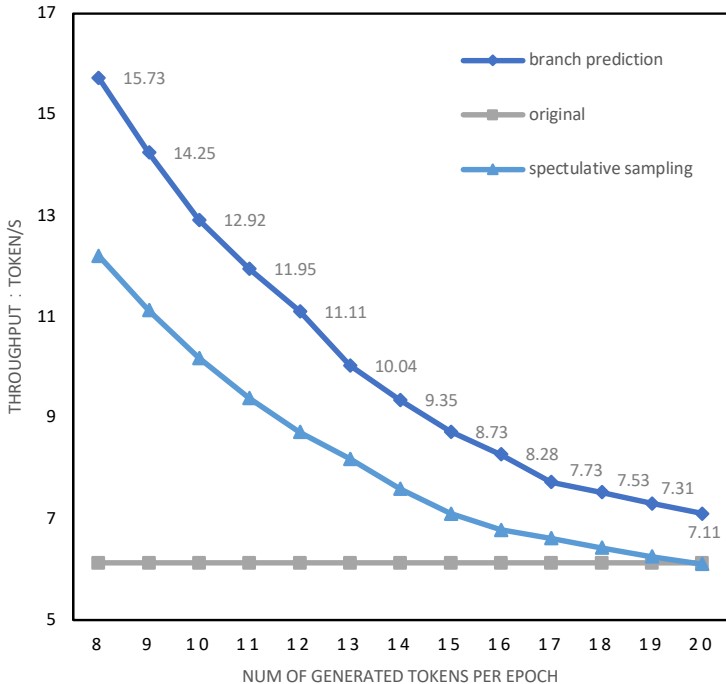

**Figure 10.** Comparison of different algorithms' throughput as the number of tokens generated by the approximate model in a single round changes. Higher is better.

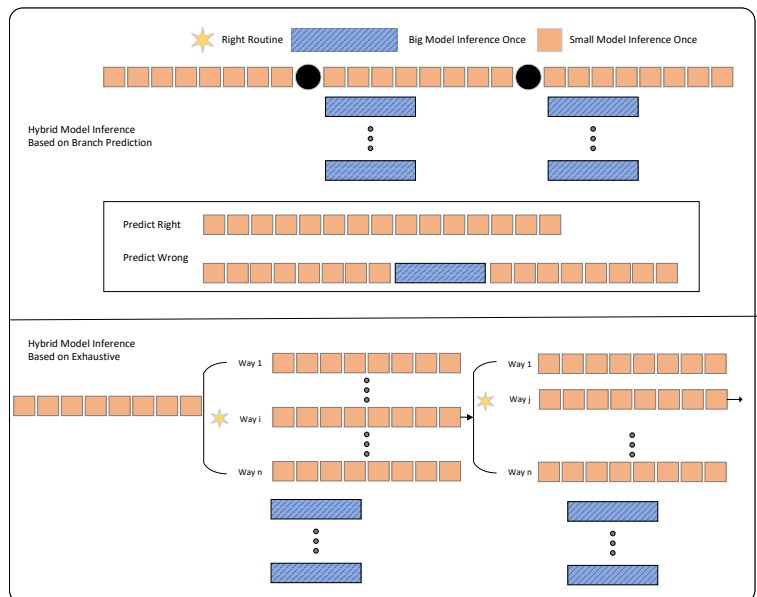

**Figure 11.** The comparison between the exhaustive method and the branch-prediction-based algorithm for hybrid model inference highlights a fundamental difference in approach. Instead of selecting a single potential starting point with the assistance of the branch-prediction function, the exhaustive method enumerates all possible starting points and performs generation in parallel. This approach significantly diminishes the time required for large model validation.

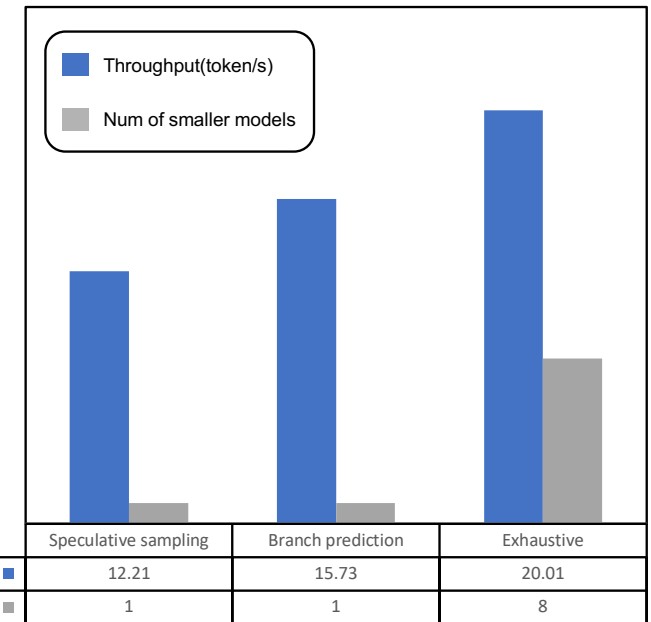

**Figure 12.** Comparative analysis of computational resource consumption between exhaustive and branch-prediction methods. This figure illustrates the comparison of computational resource consumption between the exhaustive method and branch-prediction method in text-generation tasks. The exhaustive method, achieving the highest inference speed of 20.01 tokens/s, correspondingly incurs an up to 8-fold increase in computational resource usage. The graph distinctly shows how the exhaustive method's demand for computational resources exponentially grows with an increase in n (the number of tokens generated in a single round), highlighting the importance of balancing computational resources when choosing implementation strategies.

## 7. Conclusions

This study addresses the problem of performance bottlenecks caused by increasing model sizes in large language model inference and proposes a hybrid model inference acceleration algorithm based on branch prediction. By introducing a branch-prediction scheduling strategy and designing personalized prediction functions for different model combinations based on empirical distributions and theoretical assumptions, the inference process is optimized for acceleration. The branch-prediction-based inference acceleration algorithm proposed in this paper significantly improves inference speed while aligning with the output of the target large model.

In terms of validating previous theories, this paper thoroughly analyzes existing research attempts to accelerate large model inference and chooses a hybrid model acceleration strategy to avoid the need for retraining models. It further introduces branch prediction to accelerate hybrid inference and reduce workflow congestion. The research results show that compared to the target model and the latest advanced acceleration efforts, the algorithm proposed in this paper effectively enhances the inference speed. This method solves the performance issue of the increased inference time in large language models, offering a viable solution to the challenge of maintaining real-time inference capabilities as model sizes continue to expand.

**Author Contributions:** Conceptualization, G.D. and J.C.; methodology, G.D. and Y.Z. (Yueying Zhou); software, G.D.; validation, G.D.; formal analysis, G.D.; investigation, G.D.; resources, G.D.; data curation, G.D.; writing—original draft preparation, G.D.; writing—review and editing, X.Z. and Y.Z. (Yongxin Zhu); visualization, G.D.; supervision, X.Z.; project administration, X.Z. and Y.Z. (Yongxin Zhu); funding acquisition, X.Z. and Y.Z. (Yongxin Zhu). All authors have read and agreed to the published version of the manuscript.

**Funding:** This research was supported by the National Natural Science Foundation of China under grant number 12373113, as well as the National SKA Program of China (grant no. 2020SKA0120202).

**Data Availability Statement:** The Part of this research data and model could be found in https://github.com/karpathy/llama2.c/blob/master/run.c accessed on 3 March 2024.

**Conflicts of Interest:** The authors declare no conflicts of interest. The funders had no role in the design of the study; in the collection, analyses, or interpretation of data; in the writing of the manuscript; or in the decision to publish the results.

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
