# Peer review of "Large Language Model Inference Acceleration Based on Hybrid Model Branch Prediction"

_electronics, doi:10.3390/electronics13071376_

Round 1
Reviewer 1 Report
Comments and Suggestions for Authors
The article is well written and organized. The article should include brief overview of the research methodology, research problem and also discuss the challenges that lie in a particular field.
Moreover, it should describe the pros and cons of each existing technique in the literature review section.
Highlight the benchmark datasets that have been used for the evaluation of existing techniques and compare results on specific parameters.
The present abstract is not clear. Enhance it with results, dataset, significance and contributions of the work. Provide the computing time of the experiments performed using the proposed method.
Following points are expecting from author(s) to make a quality paper.
-Re-frame the abstract with significant details.
-Provide detailed description of methodology used in the proposed model.
-How the proposed model is better than existing one that use the same datasets? Give justification by comparing with specific parameters.
-Give brief overview of benchmark datasets.
-Compared proposed result in the form of graphically to make more effective. Also provide computing time of proposed model.
-grammatical mistake is there in the paper so proof reading is required.
Specific comments to the Author
Following points are expecting from author(s) to make a quality paper.
-Re-frame the abstract with significant details.
-Provide detailed description of methodology used in the proposed model.
-How the proposed model is better than existing one that use the same datasets? Give justification by comparing with specific parameters.
-Give brief overview of benchmark datasets.
-Compared proposed result in the form of graphically to make more effective. also provide computing time of proposed model.
-grammatical mistake is there in the paper so proof reading is required.
Reviewer 2 Report
Comments and Suggestions for Authors
1. The figures could be improved. The captions aren't easy to understand. Figure 1 doesn't have an explanation, and Figure 2 isn't clear in context. The term "ideal" in the caption isn't clear, and figures like 9 and 10 lack detail because the captions or context don't explain the experiment well.
2. The method presentation needs to be more professional. Some terms aren't commonly used and need definition or explanation, such as "model draft" and "extreme computing resources." While the method is presented clearly, the explanation should be more coherent to help readers understand the pipeline and the purpose of each step.
3. The related work section needs more literature review. Including a review table would help readers understand existing research and how this work contributes to it.
4. The baseline method isn't clearly explained. It should be compared with similar methods that achieve accuracy while saving inference time.
5. The experiment setup lacks detail. More information about the specifics of training or optimization is needed. Additionally, the comparison requires clearer experiment details to ensure fairness.
Comments on the Quality of English Language
The clarity of the scientific presentation needs significant improvement. This can be achieved by providing clear definitions for terms, ensuring a coherent presentation of the material, and enhancing the overall organization.
Round 2
Reviewer 2 Report
Comments and Suggestions for Authors
The revision significantly improved the manuscript. I think the current version can be improved by minor editing for publication.
Comments on the Quality of English Language
The revised manuscript shows much more eloquent presentation. Minor editing is needed to further polish the manuscript for publication.